# Evidence of a Novel Mitochondrial Signature in Systemic Sclerosis Patients with Chronic Fatigue Syndrome

**DOI:** 10.3390/ijms241512057

**Published:** 2023-07-27

**Authors:** Charmaine van Eeden, Desiree Redmond, Naima Mohazab, Maggie J. Larché, Andrew L. Mason, Jan Willem Cohen Tervaert, Mohammed S. Osman

**Affiliations:** 1Division of Rheumatology, Department of Medicine, Faculty of Medicine and Dentistry, University of Alberta, Edmonton, AB T6G 2G3, Canada; 2Division of Rheumatology, Department of Medicine, McMaster University, Hamilton, ON L8N 3Z5, Canada; 3Division of Gastroenterology, Department of Medicine, Faculty of Medicine and Dentistry, University of Alberta, Edmonton, AB T6G 2G3, Canada

**Keywords:** systemic sclerosis, myalgic encephalomyelitis, fatigue, mitochondria

## Abstract

Symptoms of myalgic encephalomyelitis/chronic fatigue syndrome (ME/CFS) are common in rheumatic diseases, but no studies report the frequency of these in early systemic sclerosis. There are no known biomarkers that can distinguish between patients with ME/CFS, although mitochondrial abnormalities are often demonstrated. We sought to assess the prevalence of ME/CFS in limited cutaneous SSc (lcSSc) patients early in their disease (<5 years from the onset of non-Raynaud’s symptoms) and to determine if alterations in mitochondrial electron transport chain (ETC) transcripts and mitochondrial DNA (mtDNA) integrity could be used to distinguish between fatigued and non-fatigued patients. All SSc patients met ACR/EULAR classification criteria. ME/CFS-related symptoms were assessed through validated questionnaires, and the expression of ETC transcripts and mtDNA integrity were quantified via qPCR. SSc patients with ME/CFS could be distinguished from non-fatigued patients through ETC gene analysis; specifically, reduced expression of ND4 and CyB and increased expression of Cox7C. ND4 and CyB expression correlated with indicators of disease severity. Further prospective and functional studies are needed to determine if this altered signature can be further utilized to better identify ME/CFS in SSc patients, and whether ME/CFS in early SSc disease could predict more severe disease outcomes.

## 1. Introduction

Systemic sclerosis (SSc) is a heterogeneous disease characterized by immune dysregulation, vasculopathy and fibrosis of the skin and visceral organs [1]. These changes can be present early in the disease (<5 years), with patients having Raynaud’s phenomenon (RP) SSc-specific autoantibodies (e.g., anticentromere antibodies (ACA), anti-Scl70 autoantibodies) and a SSc-pattern that is detected using nailfold video capillaroscopy (NVC). The most common clinical manifestations present in early SSc include puffy fingers, digital pits/ulcers and/or interstitial lung disease [1]. Early in SSc, skin fibrosis may not be clinically apparent in many patients [1]. Many SSc patients may suffer from nonspecific symptoms such as fatigue or muscle pain, which can be mistaken for fibromyalgia (FM). These symptoms may have profound effects on the quality of life in patients with SSc, and potentially result in loss of income [2].

Fatigue in SSc is common in patients with long-standing SSc with rates ranging between 48–79% [2,3]. In many studies, it has been suggested that lung fibrosis is an important driver of fatigue, but this is not well established [2]. In contrast to symptoms of just being tired, ME/CFS is characterized by severe fatigue associated with unrefreshing sleep, post-exertional malaise, cognitive dysfunction and dysautonomia symptoms such as orthostatic intolerance. Many patients may also develop widespread pain, resulting in comorbid FM [4].

Both fatigue and FM are frequently associated with other rheumatic diseases, though studies assessing the frequency of ME/CFS and FM in established SSc are few [5,6,7]. Perrot et al. found that 27.8% of SSc patients (*n* = 122) presented with FM, and that these comorbid patients had higher BMIs, greater pain intensity and more diffuse pain [6]. In patients with long standing SSc, fatigue is associated with increased skin fibrosis, which is known to be associated with a poor outcome [2]. Importantly, no studies have quantified the frequency of myalgic encephalomyelitis/chronic fatigue syndrome (ME/CFS) and FM in SSc patients particularly early on in their course of the disease [3]

Though the etiology of ME/CFS and FM remains poorly defined, mitochondrial dysregulation may play a prominent role in their pathophysiology [8,9]. Importantly, many patients with idiopathic FM demonstrate dysregulated expression of proteins in the mitochondrial electron transport chain (ETC) (e.g., ND4 (complex I), Cytochrome B (CyB, complex III) and COX7C (complex IV)) [8]. Mitochondria are responsible for regulating metabolic activity and the generation of important intermediates, and play integral roles in critical cellular decisions such as cell death and proliferation. Notably, oxidative phosphorylation, utilizing the ETC, produces important metabolic intermediates and ATP [10].

Though the majority of mitochondrial proteins are nuclear-encoded, a subset of core ETC proteins are encoded by the mitochondrial genome (mtDNA). Hence, mitochondrial biogenesis requires coordinated expression from both genomes. ND4 and CyB are mitochondrially encoded and Cox7C is encoded by the nucleus [10]. Loss of mitochondrial functions as a result of altered ETC gene expression has been linked to aging and diseases associated with accelerated aging in the immune system, such as atherosclerosis, Alzheimer’s and multiple sclerosis [11,12,13,14,15]. Notably, oxidative stress and the disruption of oxygen homeostasis has been linked to the development of vascular remodeling and fibrosis [16,17].

Based on clinical observations in our SSc patient cohort, we hypothesized that many patients with early SSc (disease duration <5 years) may fulfill the diagnostic criteria for ME/CFS and/or FM. We postulate that markers of mitochondrial dysfunction may be preferentially dysregulated in patients with SSc suffering from ME/CFS, compared to those with SSc without ME/CFS.

## 2. Results

### 2.1. Demographic and Clinical Characteristics of Study Population

Consent for inclusion into this study was obtained from 12 fatigued and 12 non-fatigued, sequentially recruited lcSSc patients with non-Raynaud’s related symptoms <5 years that were age- and sex-matched to healthy controls. The 12 fatigued patients fulfilled the classification criteria for ME/CFS (SSc-CFS); the 12 non-fatigued patients did not (SSc-NCFS). All SSc patients had Raynaud’s phenomenon with a SSc profile on nailfold video-capillaroscopy (NVC) [18]. Median modified Rodnan skin scores (mRSS) were zero. Median forced expiratory volume in one second (FEV1)/forced vital capacity (FVC) ratios—77.5 (72.8;82.3); single-breath diffusing capacity of the lung (DLCO SB)—18.61 (15.8;22.3); FEV1—2.55 (2.1;3.0); and FVC—3.29 (2.6;3.8) values were within normal reference ranges. A total of 87.5% (21/24) of SSc patients had puffy fingers; few (25% (6/24)) had digital ulcers; 37.5% (9/24) reported arthralgia’s; and none suffered joint contractures. Gastrointestinal symptoms were reported by 29.1% (7/24) of patients, and interstitial lung disease (ILD) (confirmed by chest computer tomography) was documented in 16.6% (4/24).

Comparing SSc patients with ME/CFS (SSc-CFS), SSc without ME/CFS (SSc-NCFS) and healthy controls (HC), we found no statistical differences between age, gender, body mass index (BMI) and C-reactive protein (CRP) (Table 1). Thyroid-stimulating hormone levels were significantly higher in the SSc-CFS group (2.63) when compared to both SSc-NCFS (1.54, *p* = 0.007) and HC (1.29, *p* = 0.009), though all individuals had levels within the normal reference range (0.2–4.0 mU/L). SSc disease-related factors, including mRSS score, digital ulcers, autoantibodies, puffy fingers, lung function (DLCOSB, FEV, FVC) and ILD, were similar between SSc-CFS and SSc-NCFS patients (Table 1). Arthralgia was, however, more often present in SSc-CFS patients (X^2^ = 4.44, *p* = 0.035). Immunomodulatory and vasodilator medication use was similar between SSc-CFS and SSc-NCFS patients (X^2^ = 1.53, *p* = 0.675), though immunomodulatory therapy was associated with higher rates of arthralgia (X^2^ = 15.1, p = 0.002), and vasodilator use (without immunomodulators) associated with digital ulcers (X^2^ = 9.53, *p* = 0.025) (Appendix A).

### 2.2. ME/CFS and FM in SSc

As we predicted, severe fatigue in SSc-CFS patients (as defined by the MFI (67.5) and FACIT (24)) was significantly higher than in the SSc-NCFS group (39, *p* < 0.001, 48, *p* < 0.001) (Table 1, Appendix A). A total of 66.6% (8/12) of SSc-CFS patients also fulfilled the diagnostic criteria for FM. Pain, as measured by the widespread pain index (WPI), was significantly higher in SSc-CFS (6.5) patients than both SSc-NCFS patients (1, *p* < 0.001) and HCs (0.5, *p* < 0.001). The symptom severity score (SSS), which quantifies the severity of fatigue, tiredness and cognitive impairment in the preceding week, was higher in SSc-CFS patients (7) than both SSc-NCFS (3, *p* < 0.001) and HCs (1, *p* < 0.001) (Appendix A). Quality of life, as measured by the physical (PCS) and mental (MCS) components of the SF-36, was significantly lower in SSc-CFS patients when compared with both SSc-NCFS and HCs (*p* < 0.001) (Table 1).

Rates of depression were similar in all patient and control groups (Table 1), though HCs (0.5, *p* < 0.001) and SSc-NCFS (1.5, *p* = 0.002) patients had significantly lower HADS_depression scores than SSc-CFS patients (5) (Appendix A). Rates of anxiety, as well as HADS_anxiety scores, were significantly higher in SSc-CFS, compared to both SSc-NCFS and HCs (*p* < 0.050) (Table 1, Appendix A). Similar rates of sleep disturbance were seen for SSc-CFS patients and SSc-NCFS patients (X^2^ = 2.25, *p* = 0.133) (Table 1), but SSc-CFS patients had significantly higher Pittsburgh Sleep Quality Index (PSQI) scores (13) than SSc-NCFS patients (3.5, *p* < 0.001) (Appendix A). Cognitive failure was similar between all groups (Table 1, Appendix A). Immunomodulatory and vasodilator medication use was not associated with symptoms related to ME/CFS in our SSc patients (Appendix A).

### 2.3. ETC Differential Expression and Cell Free Mitochondrial DNA Integrity

We investigated expression levels of genes within the ETC that have previously been associated with altered mitochondrial functions in ME/CFS and/or FM, using whole-blood RNA purified from PAX gene extractions. Compared to SSc-NCFS patients (0.47), SSc-CFS patients showed a trend of reduced expression of ND4 (0.34, *p* = 0.051) (Figure 1A). CyB expression was significantly lower for SSc-CFS (0.21) when compared to both HCs (0.42, *p* = 0.030) and SSc-NCFS patients (0.38, *p* = 0.044) (Figure 1A). Intriguingly, we also saw increased expression of Cox7C in SSc-CFS (−0.86) when compared to SSc-NCFS (−1.11, *p* = 0.031) (Figure 1A). This was not associated with differences in nuclear-encoded CoxIV protein levels, between SSc-CFS and SSc-NCFS patients (Appendix A). Dloop expression as a measure of mtDNA transcription, was not different between the groups. Additionally, protein levels of TFAM were similar between SSc-CFS and SSc-NCFS patients (Appendix A). TFAM is a core mitochondrial transcription factor, known to directly control mtDNA copy number [19]. We did not appreciate any significant differences in mtDNA integrity between SSc-CFS (1.03), SSc-NCFS (1.04, *p* = 0.137) and HCs (1.05, *p* = 0.251), though SSc-CFS patients appeared to have lower levels (Figure 1B).

We individually assessed the correlates of ETC gene expression and mtDNA integrity in our SSc patients. We found that both CyB and Cox7C correlated with fatigue as measured by the FACIT (0.40, *p* = 0.057; −0.46, *p* = 0.030), whilst mtDNA integrity correlated with fatigue as measured by the MFI (−0.59, *p* = 0.003) (Appendix A). Both CyB expression and mtDNA integrity correlated with reduced quality of life as measured by SF36-Vitality (0.40, *p* = 0.052; 0.49, *p* = 0.018) (Appendix A). A noteworthy finding was the correlation of both ND4 and CyB expression with disease-related factors, specifically increased skin involvement (mRSS) (−0.41, *p* = 0.045; −0.45, *p* = 0.024) and reduced lung diffusion capacity (DLCO SB) (0.48, *p* = 0.018; 0.49, *p* = 0.015). ND4 expression was also significantly correlated with FVC and FEV_1_ values (0.46, *p* = 0.024; 0.46, *p* = 0.022) (Appendix A).

## 3. Discussion

We show for the first time that many patients with early SSc (disease duration <5 years) fulfill the classification criteria for ME/CFS and FM. Remarkably, the rate of both ME/CFS and FM in these patients is consistent with values described in long-standing SSc [2]. As previously documented in fatigued patients with long-standing SSc, we found that early SSc disease patients had significantly higher scores with regards to fibromyalgia symptoms (WPI and SSS), sleep disturbances, anxiety and depression [2]. SSc-CFS patients could not be distinguished from SSc-NCFS patients with regards to disease-related factors such as skin thickening, lung function, ILD, CRP, digital ulcers and gastrointestinal disturbances.

Though in the normal range, TSH levels were shown to be higher in SSc-CFS patients; this is in keeping with previous studies, which have shown a significant increase in TSH levels in SSc patients compared to healthy controls, with mean values staying within normal ranges [19]. Increased TSH levels have, however, been shown to correlate with SSc disease severity, as indicated by decreased FEV1 and FVC values [20]. Rates of arthralgia were higher in SSc-CFS patients, which is consistent with symptoms of fibromyalgia, and possibly the presence of an interferon signature [21]. Of interest is the finding that a trend was evident for the reduced presence of anticentromere antibodies (ACA) in SSc-CFS patients. These antibodies are useful prognostic markers in SSc disease progression, where their presence is associated with a less fibrotic disease profile [22] and a reduced risk of other features such as malignancy [23]. Additionally, ACAs are described as being protective against scleroderma renal crisis and interstitial lung disease, even in patients with diffuse SSc [24]. Another important finding was that the use of immunomodulator and vasodilator medications did not impact the development of fatigue in our patients. Though these therapies were respectively associated with arthralgia and digital ulcers, this is likely due to treatment bias.

Looking at potential markers of mitochondrial dysregulation, we found that mitochondrially encoded genes ND4 and CyB had reduced expression in SSc-CFS patients, whilst nuclear gene Cox7C had elevated expression. Increased levels of Cox7C (without changes in CoxIV) is intriguing, since this suggests that Cox7C may be providing a compensatory role in patients with ME/CFS. In addition to stabilizing Complex IV, Cox7C has also recently been implicated in promoting more efficient energy utilization via respirasome supercomplexes [25]. Supercomplexes may play a role in stabilizing individual ETC complexes, reducing reactive oxygen species (ROS) production and increasing ETC efficiency [25].

Mitochondrial suppression is recognized as an early (and required) feature of immune cell activation [26]. It is also established that loss of mitochondrial ETC transcripts (particularly in ETC Complex I) may promote various inflammatory diseases [8]. Fibrosis in SSc is associated with metabolic dysregulation [27]. In particular, Zhou et al. showed that impaired TFAM expression led to mitochondrial damage, reduced OXPHOS capacity and transcriptional changes that drive tissue fibrosis in SSc patients [28]. Similarly, preclinical animal models of SSc with altered mitochondrial functions develop more severe fibrotic complications [29]. Furthermore, patients with diffuse SSc have alterations in molecules (such as CD38), which may result in more suppressed mitochondrial functions [30].

An intriguing finding with regards to our ETC gene expression was that increased age, BMI and CRP did not seem to play a factor in this altered gene signature. In particular, obesity has been linked to decreased mitochondrial gene expression in adipose tissues [25], and the majority of our SSc patients were overweight. We instead found that reduced ND4 and CyB gene expression was correlated with increased skin involvement and reduced DLCO SB, and that ND4 expression was correlated with reduced FVC and FEV_1_ values. DLCO SB, FVC and FEV_1_ are considered surrogate markers for SSc disease progression and the development of ILD, which is the leading cause of mortality in SSc patients [26].

We postulate that the presence of ME/CFS with associated mitochondrial dysfunction in lcSSc patients with less than 5 years of disease may be associated with increased risk of developing more severe disease-specific complications in the future. The altered ETC gene signature in these patients may also hint at the mechanisms underlying disease progression. Future large prospective studies assessing these patients and disease-associated complications over time are required to verify this hypothesis.

The heterogeneity of ME/CFS has made diagnosis and treatment of these patients challenging. Conducting clinical trials for these patients has been most challenging, with inconsistent conclusions likely stemming from patient heterogeneity. Hence, biomarkers that can strategically identify patients more harmoniously may inform future clinical trials. It is well established that fatigue in rheumatic diseases such as SSc can be explained, at least in part, by their disease. In our SSc cohort of patients early in their disease, clinical variables associated with SSc could not distinguish SSc-CFS patients from SSc-NCFS patients. Both fatigue and mitochondrial dysfunction have, however, been linked to more severe SSc outcomes [27,28]. In our study, a simple assay measuring differences in ETC gene expression was able to distinguish these two groups (Figure 2). Additionally, these changes in ETC gene expression also showed a connection with increased skin and lung function measures.

There are several limitations to our study. These include the small sample size and single-center design. Our exploratory results will need to be replicated and confirmed in larger, more ethnically diverse cohorts. Though we did not directly measure changes of electron transport chain activity in cells, other groups assessing patients with ME/CFS have [9,29]. The observed gene expression changes will thus require follow-up in functional assays to further elucidate the possible mechanisms affecting mitochondrial function in SSc patients who meet the criteria for ME/CFS.

## 4. Materials and Methods

### 4.1. Participants

Participants included in our study were referred to the Rheumatology Clinics at the University of Alberta. The study was approved by the University of Alberta Research Ethics Board (Pro00085583, Pro00090050), and all research was performed in accordance with relevant guidelines. All methods were performed in accordance with the Declaration of Helsinki. Patients were recruited to the study during their scheduled rheumatology appointments. Written informed consent was obtained from all the study participants. Patients were not involved in setting the research question or outcome measures.

All of our SSc patients meeting the 2013 European Alliance of Associations for Rheumatology/American College of Rheumatology (EULAR/ACR) classification criteria for SSc were invited to take part [30]. All patients were tested for antinuclear antibodies and SSc-specific autoantibodies (Appendix A) [31] in a reference clinical laboratory (Mitogen^Dx^ laboratories, Calgary, Alberta). Patients underwent simultaneous clinical and nailfold capillaroscopy (NVC) assessments at the time of enrollment [32]. All patients with SSc had an SSc-spectrum capillaroscopy pattern [1,18]. Consecutive SSc patients who had non-Raynaud’s related symptoms (Autoantibodies, positive NVC) for less that <5 years, who gave informed consent, participated in this study. Patients were screened for the use of both immunomodulatory (Hydroxychloroquine, Methotrexate, Folate, Mycophenolate, Prednisone) and vasodilator medications (Amlodipine, Nifedipine, Sildenafil, Tadalafil). None of the patients included were treated with rituximab or tocilizumab.

FM was diagnosed using the modified 2010 and 2016 American College of Rheumatology classification criteria for fibromyalgia [33,34]. Diagnosis of ME/CFS was based on both the International and Canadian Consensus Criteria [35,36,37]. Clinical parameters such as BMI, and laboratory parameters such C-reactive protein (CRP) and thyroid-stimulating hormone (TSH) levels were also collected. Healthy controls (HC)s were volunteers that did not suffer from a rheumatic disease, hypothyroidism or diabetes requiring insulin. Similarly, patients with idiopathic FM who visited our rheumatology clinic and who met the FM 2010 and 2016 classification criteria [33,34] and did not suffer from an underlying inflammatory condition, hypothyroidism and/or diabetes were included in our study.

### 4.2. Questionnaires

Enrolled patients were assessed for the presence of ME/CFS and/or FM using validated questionnaires (e.g., DePaul symptom questionnaire (DSQ-2), wide-scale pain index (WPI) and symptom severity scale (SSS), and the 36-Item Short Form Survey (SF-36)) (Appendix A). Comorbidities (e.g., anxiety, sleep disturbances and depression) were also assessed using validated questionnaires (Appendix A).

### 4.3. Mitochondrial Electron Transport Chain Expression

We selected 12 sequential patients from each of our SSc groups (SSc with ME/CFS (SSc-CFS) and SSc without ME/CFS (SSc-NCFS)) and 9 patients for the healthy and idiopathic FM controls for mitochondrial gene expression analyses. Blood-derived mRNA was extracted from PAXgene tubes using the PAXgene Blood RNA Kit (PreAnalytiX). Subsequently, real-time PCR was carried out using Taqman Fast Virus 1 step Master Mix (Applied Biosystems) and D-loop (MT-7S), ND4, MT-CYB, COX7C, β-Actin and GAPDH expression kits (Hs02596861_s1, Hs02596876_g1, Hs02596867_s1, Hs01595220_g1, Hs99999903_m1, Hs99999905_m1). Gene expression was normalized to GAPDH. Expression was calculated as the log(10) of ddCT. PCRs were conducted in duplicate, with three replicates and the three means averaged. The median log(ddCT) values were determined for each patient group and visualized as a violin plot.

### 4.4. Cell-Free Mitochondrial DNA Analysis

Cell-free DNA was extracted from 500 μL serum using the QIAamp UltraSens Virus Kit (QIAGEN), according to the manufacturer’s instructions. Quantitative analysis of mtDNA in serum was performed by SYBR Green quantitative real-time polymerase chain reaction (PCR). Two previously published primer sets were used to analyze the levels of a 79 bp fragment of the mitochondrial 16S-RNA (corresponding to DNA released by apoptotic cells) and 230 bp fragment (corresponding to mtDNA released by necrosis), then normalized to cell-free GAPDH DNA from the same sample, as previously described [38,39] (Appendix A). Mitochondrial fragmentation was defined as mtDNA integrity, which was calculated as the relative amount of mtDNA230 to mtDNA79. Each 10 μL reaction consisted of 5 μL Fast SYBR Green master mix (Applied Biosystems), 1 μL DNA and 0.5 μL of forward and reverse primer (8000 nM). PCR conditions were as follows: 95 °C for 15 min, followed by 40 cycles of 95 °C for 15 s, 60 °C for 30 s, 72 °C for 30 s. Melt curve analysis was applied to confirm specificity, at 95 °C for 15 s, 60 °C for 15 s and 95 °C for 15 s. PCRs were conducted in triplicate, with the mean of two replicate runs used for analysis.

### 4.5. Statistical Analysis

All statistical analyses were completed using STATA 17 (StataCorp). For continuous variables, median and interquartile range are indicated with the exact *p*-value, as determined by the Wilcoxon rank-sum test. For categorical variables, chi-square analysis was carried out with Fisher’s exact test. Variables are indicated as the total number of positive responses and the percentage of positive responses within the population. Spearman’s rank-order was used for correlation coefficient analysis.

## 5. Conclusions

ME/CFS plays a pivotal role in early morbidity in a subset of SSc patients. We identified a mitochondrial signature in SSc patients with ME/CFS, which distinguishes them from SSc patients without this comorbidity. Gene expression changes in mitochondrially encoded ND4 and CyB also correlated with measures of disease severity. Further investigations into the functional consequences of the observed mitochondrial changes may lead to the facilitation of therapeutic intervention studies and possible treatment options, a current unmet need in SSc patients with ME/CFS. Together, our data suggest that the presence of ME/CFS in SSc reflects an independent manifestation of SSc, which we hypothesize may be linked to poor disease-related outcomes. Future studies assessing this possibility in SSc and possibly in other rheumatic diseases should be tested.

## Figures and Tables

**Figure 1 ijms-24-12057-f001:**
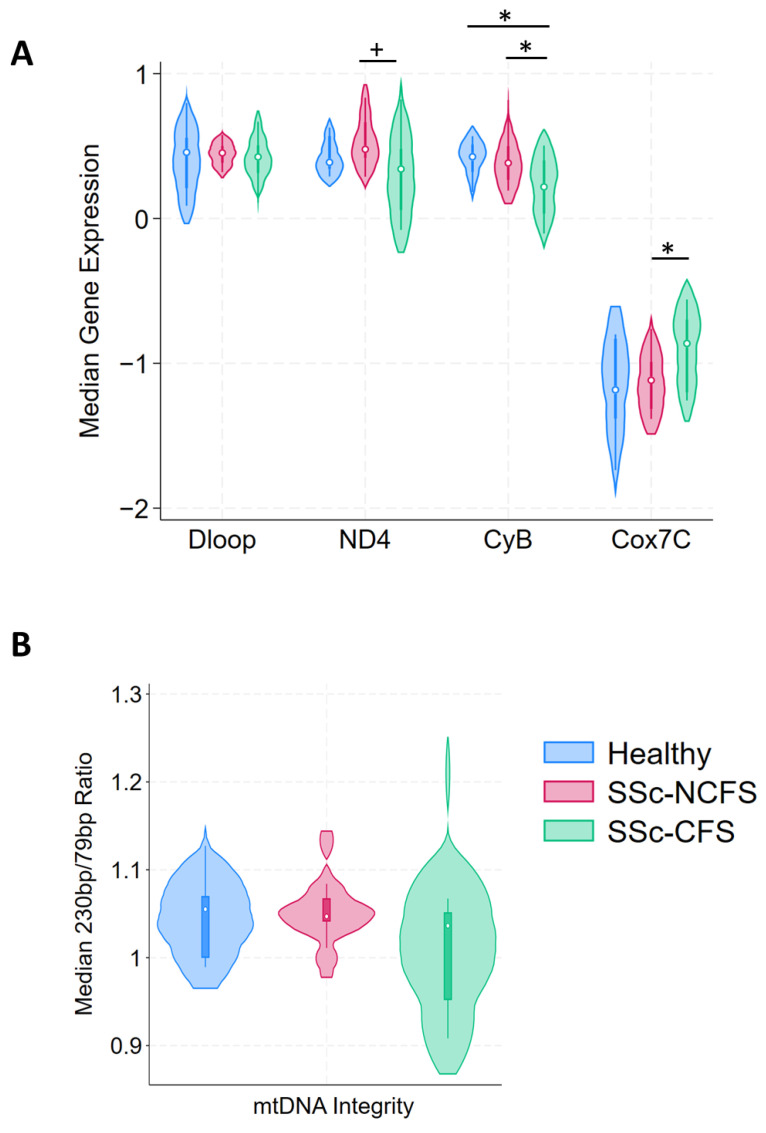
Mitochondrial electron transport chain gene expression and mitochondrial cell-free DNA. Mitochondrially encoded ND4 and CyB gene expression was reduced in SSc-CFS patients, compared to SSc-NCFS patients. Nuclear-encoded Cox7C expression was significantly higher in SSc-CFS patients. Mitochondrial transcription, as indicated by Dloop expression, was not significantly different between disease groups (**A**). Cell-free mitochondrial DNA integrity was not significantly different between each group (**B**). mtDNA integrity refers to the ratio of mtDNA230 to mtDNA79 fragments. Median and interquartile range are indicated. (*p*-values: + = 0.050, * <0.050).

**Figure 2 ijms-24-12057-f002:**
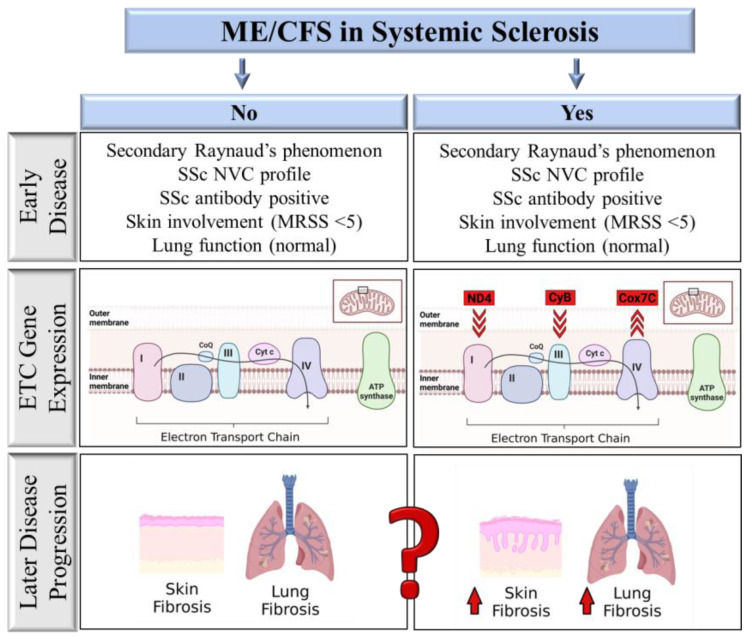
Comparison of factors associated with ME/CFS in SSc patients. SSc-CFS and SSc-NCFS patients are not distinguishable from each other based on early SSc disease-related factors. Electron transport chain gene (ETC) expression distinguishes SSc-CFS patients from SSc-NCFS patients, with reduced ND4 and CyB expression and increased Cox7C gene expression. ETC gene expression correlates with measures of disease severity, including skin thickening and lung function.

**Table 1 ijms-24-12057-t001:** Patient demographics and clinical findings. SSc-CFS patients are compared to SSc-NCFS patients and healthy controls.

	SSc-CFS (*n* = 12)	SSc-NCFS(*n* = 12)	*p*	Healthy (*n* = 10)	*p*
Continuous variables	Median (IQR)	Median (IQR)		Median (IQR)	
Age	47.5 (46; 57)	53 (35.5; 63)	0.920	46 (36; 51)	0.389
BMI	30.4 (25.3; 36.0)	27.8 (22.9; 32.9)	0.178	22.5 (21.4; 34.2)	0.095
CRP (mg/L)	3.5 (2.3; 6.5)	2.7 (0.9; 7.9)	0.579	0.6 (0.5; 4.2)	0.146
TSH (mU/L)	2.63 (1.8; 3.4)	1.54 (0.83; 2.15)	0.026	1.29 (0.81; 1.47)	0.009
Lung function					
DLCO SB (ml/(min × mmHg))	17.2 (14.2; 22.7)	19.5 (16.8; 21.5)	0.265	-	-
FEV (L)	2.55 (2.0; 2.8)	2.56 (2.1; 3.1)	0.580	-	-
FVC (L)	3.18 (2.6; 3.8)	3.32 (3.0; 3.8)	0.580	-	-
mRSS	0 (0; 1.5)	0 (0; 0)	0.931	-	-
SF-36 PCS	31.25 (23.7; 44.0)	84.6 (80.3; 91.2)	<0.001	96.8 (93.7; 97.5)	<0.001
SF-36 MCS	61.25 (35.7; 70.8)	88.5 (70.6; 92.5)	0.001	90.1 (88.2; 92	<0.001
Cognitive Failure	42.5 (18.5; 51)	22 (16; 31)	0.061	26.5 (12; 38)	0.185
MFI Score	67.5 (60; 73.5.)	39 (31; 43)	<0.001	28.5 (26; 34)	<0.001
FACIT Score	24 (20.2; 29.5)	48 (44; 49.8)	<0.001	50 (50; 52)	<0.001
Categorical variables	Yes (%)	Yes (%)		Yes (%)	
Gender (female)	11/12 (91.6)	10/12 (83.3)	0.537	1/10 (10.0)	0.892
Autoantibodies					
ANA	10/12 (83.3)	11/12 (91.6)	0.537	-	-
SSc-specific autoantibodies	9/12 (75.0)	10/12 (83.3)	0.615	-	-
ACA	2/12 (16.6)	6/12 (50.0)	0.083	-	-
Medications					
Immunomodulators	3/12 (25.0)	2/12 (16.6)			
Vasodilators	3/12 (25.0)	5/12 (41.6)			
Immunomodulators and vasodilators	5/12 (41.6)	3/12 (25.0)			
No immunomodulators or vasodilators	1/12 (8.33)	2/12 (16.6)	0.675		
ILD	2/12 (16.6)	2/12 (16.6)	1.000	-	-
Digital ulcers	2/12 (16.6)	4/12 (33.3)	0.346	-	-
Arthralgia	7/12 (58.3)	2/12 (16.6)	0.035	-	-
GERD	5/12 (41.6)	2/12 (16.6)	0.178	-	-
Sleep disturbances	8/11 (72.7)	5/12 (41.6)	0.133	2/10 (20.0)	0.016
Fibromyalgia	0/12 (0.0)	8/12 (66.6)	0.001	0/10 (0.0)	0.001
HADS depression	0/12 (0.0)	2/12 (16.6)	0.140	0/10 (0.0)	0.176
HADS anxiety	0/12 (0.0)	4/12 (33.3)	0.028	0/10 (0.0)	0.044

Abbreviations: SSc-CFS: systemic sclerosis with ME/CFS; SSc-NCFS: scleroderma without ME/CFS; FM: fibromyalgia; IQR: interquartile range; BMI: body mass index; CRP: C reactive protein; mRSS: modified Rodnan skin score, TSH: thyroid-stimulating hormone; ILD: interstitial lung disease; GERD: gastroesophageal reflux disease ME/CFS: myalgic encephalomyelitis/chronic fatigue syndrome; SF-36: 36-item short form health survey; HADS: hospital anxiety and depression scale; MFI: multidimensional fatigue inventory; FACIT: functional assessment of chronic illness therapy; FEV1/FVC: first second of forced expiration to the full, forced vital capacity; DLCO: diffusing capacity of the lung for CO; DLCO/VA: transfer coefficient for the diffusion of CO into the blood; ANA: antinuclear antibody; Scl: scleroderma; ACA: anticentromere antibody; *p*: *p* value. Grey column highlights the SSc-CFS group, which is compared to both the SSc-NCFS and Healthy controls groups.

## Data Availability

The datasets generated during the current study are not publicly available due to privacy/ethical issues but are available from the corresponding author on reasonable request.

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
