# Peer review of "Evidence of a Novel Mitochondrial Signature in Systemic Sclerosis Patients with Chronic Fatigue Syndrome"

_ijms, 2023, doi:10.3390/ijms241512057_

Round 1
Reviewer 1 Report
In this article, Eeden et al reported the evidence of a novel mitochondrial signature in systemic sclerosis (SSc) patients with chronic fatigue syndrome. They have identified a mitochondrial signature in SSc patients with myalgic encephalomyelitis/chronic fatigue syndrome (ME/CFS), which distinguishes them from SSc patients without this comorbidity. Gene expression changes in mitochondrially encoded ND4 and CyB also correlated with measures of disease severity. Further investigations into the functional consequences of the observed mitochondrial changes, may lead to the facilitation of therapeutic intervention studies and possible treatment options, a current unmet need in SSc patients with ME/CFS. They suggest that the presence of ME/CFS in SSc reflects an independent manifestation of SSc, which they hypothesize may be linked to poor disease-related outcomes. They show for the first time that many patients with early SSc (disease duration <5 years), fulfill the classification criteria for ME/CFS and FM. Remarkably, the rate of both ME/CFS and FM in these patients is consistent with values described in long standing SSc. As previously documented in fatigued patients with long standing SSc, they found that early SSc disease patients had significantly higher scores with regards to fibromyalgia symptoms (WPI and SSS), sleep disturbances, anxiety and depression. SSc-CFS patients could not be distinguished from SSc-NCFS patients with regards to disease related factors such as skin thickening, lung function, ILD, CRP, digital ulcers and gastrointestinal disturbances. This is a very interesting data because of its new focus and analysis from a viewpoint that has not been reported before. I have some questions.
major concerns)
1) If the patient is being treated with immunosuppressive therapy such as steroids, immunosuppressive drugs, or vasodilators, it may affect each clinical finding. Please describe the concomitant use of each drug.
Also, please describe any differences between patients with and without these treatments, even if it is only supplemental materials.
2) Do these clinical scores and other data change with treatment? If you have data in this regard, please present the data.
Reviewer 2 Report
The authors wrote an interesting article on the prevalence of ME/CFS in limited cutaneous SSc (lcSSc) patients early in their disease and intend to determine if alterations in mitochondrial electron transport chain (ETC) transcripts and mitochondrial DNA (mtDNA) integrity could be used to distinguish between fatigued and non-fatigued patients.
However, corrections are necessary before further consideration:
All tables should have p-values with the same number of decimal places; now the authors have mixed 2 and 3 decimal places; please decide and change this also in tables and text.
All figures, including supplementary figures, need to have the y-axis determined on graphs.
One of the main problems is not including correlation factors in the Results part of the article and df for Chi-quadrat test, not only p-values.
All figures, including supplemental figures, need to have higher resolution, especially Figure 1, Figure s2, and the part of the Figure 2 that contains ECT mRNA Gene Expression. If you show this part as a separate figure, it might be easier to see the small parts. If the figure is from another author, this should be mentioned and asked for permission.
All questionnaires (lines 41-47) should be explained separately, their validity, reliability, permission to use, whether they are self-reported or not, and an explanation of the whole protocol and final scoring.
Round 2
Reviewer 2 Report
The authors have addressed a majority of asked issues. However, I am confused because the authors have left the supplementary file unchanged (they have not uploaded the new modified supplementary figures and the table S1), and the methods part with the questionnaires is also unchanged (present only in the cover letter).
The new Table S1 is not included in either the supplemental files or the answers. Table S1 still has p values with mixed 2 or 3 decimal places. The authors only stated that they changed the p-values. I cannot see this change!
Figure S1: The Y-axes are still not defined.
Figure S3 does not exist at all. You cannot change only one part of the figure and leave another unchanged with the answer: "This figure was split into two and the text was enlarged. Figure s2 and Figure s3 now have better definition. Figure s2 is shown as an example."
You need to change everything and then insert all the necessary changes into the main text (the Manuscript ), including the new text, tables, figures, etc., and highlight them in color, and also upload the new supplementary file with all the changed tables and figures into the system.
English needs some editing.
Round 3
Reviewer 2 Report
All queries addressed.